# Does nicotine exposure during adolescence modify the course of schizophrenia-like symptoms? Behavioral analysis in a phencyclidine-induced mice model

Ana Carolina Dutra-Tavares[1]☯, Alex C. Manhães[1]☯, Keila A. Semeão[1], Julyana G. Maia[1], Luciana A. Couto[1], Claudio C. Filgueiras[1], Anderson Ribeiro-Carvalho[2], Yael Abreu-Villaça[1]*

1 Departamento de Ciências Fisiológicas, Laboratório de Neurofisiologia, Instituto de Biologia Roberto Alcantara Gomes, Universidade do Estado do Rio de Janeiro (UERJ), Rio de Janeiro, RJ, Brazil, 2 Departamento de Ciências, Faculdade de Formação de Professores da Universidade do Estado do Rio de Janeiro, São Gonçalo, RJ, Brazil

☯ These authors contributed equally to this work.
* yael_a_v@yahoo.com.br, yael_a_v@pq.cnpq.br

**Data Availability Statement:** All relevant data are within the manuscript and its Supporting Information files.

## Abstract

The first symptoms of schizophrenia (SCHZ) are usually observed during adolescence, a developmental period during which first exposure to psychoactive drugs also occurs. These epidemiological findings point to adolescence as critical for nicotine addiction and SCHZ comorbidity, however it is not clear whether exposure to nicotine during this period has a detrimental impact on the development of SCHZ symptoms since there is a lack of studies that investigate the interactions between these conditions during this period of development. To elucidate the impact of a short course of nicotine exposure across the spectrum of SCHZ-like symptoms, we used a phencyclidine-induced adolescent mice model of SCHZ (2.5mg/Kg, s.c., daily, postnatal day (PN) 38-PN52; 10mg/Kg on PN53), combined with an established model of nicotine minipump infusions (24mg/Kg/day, PN37-44). Behavioral assessment began 4 days after the end of nicotine exposure (PN48) using the following tests: open field to assess the hyperlocomotion phenotype; novel object recognition, a declarative memory task; three-chamber sociability, to verify social interaction and prepulse inhibition, a measure of sensorimotor gating. Phencyclidine exposure evoked deficits in all analyzed behaviors. Nicotine history reduced the magnitude of phencyclidine-evoked hyper-locomotion and impeded the development of locomotor sensitization. It also mitigated the deficient sociability elicited by phencyclidine. In contrast, memory and sensorimotor gating deficits evoked by phencyclidine were neither improved nor worsened by nicotine history. In conclusion, our results show for the first time that nicotine history, restricted to a short period during adolescence, does not worsen SCHZ-like symptoms evoked by a phencyclidine-induced mice model.

**Funding:** This work was supported by fellowships from Conselho Nacional de Desenvolvimento Científico e Tecnológico (CNPq-BRAZIL - 132524/ 2019-8 to KAS and 141270/2019-5 to LAC) and Coordenação de Aperfeiçoamento de Pessoal de Nível Superior (CAPES-BRAZIL - finance code 001 to JGM) and grants from Fundação Carlos Chagas Filho de Amparo à Pesquisa do Estado do Rio de Janeiro (FAPERJ-BRAZIL - SEI-260003/001135/ 2020, E-26/202.792/2017 and E-26/010.100961/ 2018 to YA-V; E-26/202.266/2018 to ACD-T) and CNPq-BRAZIL (304332/2016-0 to YA-V). The funders had no role in study design, data collection and analysis, decision to publish, or preparation of the manuscript.

**Competing interests:** The authors have declared that no competing interests exist.

## 1. Introduction

About 24.9% of the world population aged 15 and over were smokers in 2015 and there is an expectation of reduction to 20.9% in 2025 [1]. These numbers notwithstanding, smoking prevalence remains disproportionately high, especially among people with a mental health condition when compared to the general population [2]. In fact, individuals with mental illness comprise more than half of nicotine-dependent smokers [3]. Among mental illnesses, schizophrenia (SCHZ) is a strong risk factor for nicotine addiction: The prevalence of conventional tobacco use in SCHZ patients is 50–90%, being 2-4× higher than the prevalence in individuals without a mental health condition [4,5]. Smoking rates are higher in SCHZ patients even when compared to patients with other mental illnesses such as mood and organic mental disorders [6]. Nicotine addiction is also more severe in SCHZ patients, as they smoke more cigarettes, extract more nicotine and carbon monoxide from each cigarette, and have greater reinforcement in the act of smoking when compared to smokers without this mental health condition [7,8]. These data are consistent with evidence that smoking is the single largest contributor to SCHZ patients reduced life expectancy [9–11].

Most SCHZ patients are lifetime smokers and were already smokers at first-admission diagnosis of psychosis [12]. Indeed, the association between SCHZ and nicotine addiction is frequently established early, still during adolescence [13]. In this regard, while the diagnosis of SCHZ usually occurs at adulthood, patients already display attenuated positive, cognitive, and negative symptoms during adolescence, in the prodromal stage of the disorder [14–16]. Adolescence is also when smoking typically begins and, even though the consumption of combustible cigarettes is progressively declining worldwide, there has been an exponential increase in electronic nicotine delivery systems (ENDS) use, particularly in adolescents [17–19]. These findings point to adolescence as a period of susceptibility to both nicotine addiction and SCHZ.

Neurobiological maturation events that underlie adolescent increased susceptibility to nicotine may also be relevant to SCHZ symptomatology. Particularly, the mesocorticolimbic pathway, involved in rewarding and reinforcing effects of nicotine [20,21] and also in the etiology of SCHZ [22,23], is still undergoing maturation processes during adolescence. The ventral tegmental area (VTA) of adolescents is functionally hyperactive. Indeed, both the expression of dopaminergic receptors and basal dopamine levels in the nucleus accumbens increase during this period of development [20]. This high mesolimbic activation contrasts with less activation of the prefrontal cortex and brings to light an imbalance associated with nonlinear rates of brain development between subcortical and frontal cortical regions [24]. While this imbalance has been proposed to underlie adolescents' suboptimal choices and actions that might lead to drug abuse [24], in SCHZ, mesocortical hypofrontality is implicated in cognitive deficits and negative symptoms, and the hyperactive mesolimbic pathway is associated with the positive symptomatology [25–27]. Among shared targets that are potential substrates for interactions between these diseases, the cholinergic system stands out [28]. Nicotine binding to presynaptic nicotinic acetylcholine receptors (nAChRs) facilitates the release of neurotransmitters such as dopamine, glutamate, serotonin and gamma aminobutyric acid (GABA) [29,30], all of which are involved in SCHZ etiology [31,32]. During adolescence, there is a peak of $\alpha 4^*$ nAChRs in VTA dopaminergic neurons [33] and robust and persistent nicotine-evoked nAChRs upregulation in the midbrain, hippocampus and cerebral cortex [34,35]. Studies that used a SCHZ model produced by neonatal ventral hippocampal lesions showed reduced nAChR binding in the frontal cortex of SCHZ adolescent rats, and nAChR upregulation in response to previous nicotine exposure [36].

The aforementioned findings not only pose adolescence as a period when the brain shows transitional changes that might be relevant to both nicotine addiction and SCHZ, but also suggest that the strong and complex interaction between these diseases is already evident during this period of brain development. Given the peculiarities of the adolescent brain, it is reasonable to suppose that even a short period of nicotine exposure during adolescence changes the course of SCHZ. Despite that, to date, only a handful of studies in human subjects cared to focus on this period of comorbid susceptibility. Ethical concerns, the difficulty in finding a suitable control group [6] and insufficient control of confounding factors [37] may have hampered advances in this area, and these difficulties are most likely more accentuated when it comes to underage subjects.

In this regard, animal models, which have been used for decades to investigate the neural bases of both nicotine addiction and SCHZ and, more recently, the interactions between these diseases, may be particularly useful. Regarding nicotine, the use of subcutaneous osmotic minipumps allows for a constant dose, a controlled period of administration and has been used as a surrogate for tobacco smoking [38,39]. Repeated administration of rodents with phencyclidine, a non-competitive NMDA receptor antagonist, is considered the best validated and most widely used model in SCHZ research [40,41]. It produces behavioral alterations associated with the three core (positive, negative and cognitive) symptoms of SCHZ [42]; reduces prepulse inhibition (PPI), a SCHZ endophenotype [43,44] and evokes neurochemical abnormalities associated with SCHZ such as hyperactivity of the mesolimbic dopaminergic pathway and hypoactivity of the mesocortical one [42,45]. Particularly relevant to the current study, the dysregulation of the glutamatergic system has been suggested to play a role in the progression of SCHZ from the prodromal phase to psychosis onset [46–48].

Here, we used mice models to elucidate the impact of a short course of adolescent nicotine exposure across the spectrum of SCHZ-like symptoms. Phencyclidine administered on a daily basis guaranteed the development of deficits associated with SCHZ. As for nicotine, exposure was restricted to a short period so that the impact of nicotine history on SCHZ symptomatology could be evaluated. Overall, examining the individual effects of phencyclidine and adolescent nicotine history, as well as the interactions between these insults, will preclinically delineate factors involved in the SCHZ and nicotine addiction comorbidity during the early course of these diseases. This knowledge has broad potential benefits: 1) to adjust anti-smoking policies or to develop new ones to effectively assist young SCHZ patients, 2) to instruct health professionals on the management of young SCHZ patients that smoke; 3) to open new possibilities for the investigation of therapeutic approaches that take into account peculiarities of adolescence to help SCHZ patients have a better quality of life.

## 2. Methods

### 2.1. Materials

Nicotine *free base* was purchased from Sigma Chemical Co. (St. Louis, MO, USA). Phencyclidine was purchased from Alomone Labs (Jerusalem, Israel). VETEC Química Fina Ltda (Rio de Janeiro, RJ, Brazil) was the source for all other reagents.

### 2.2. Animals and treatment (Fig 1)

All experimental procedures were approved by the Institute of Biology/UERJ Ethical Committee for Animal Research (protocol#: CEUA/033/2018), minimizing the number of animals used and avoiding animal suffering, in accordance with Brazilian Law # 11.794/2008 and with the National Institutes of Health guide for the care and use of Laboratory animals (NIH Publications No. 8023, revised 1978). All mice were kept in our animal facility housed in groups of

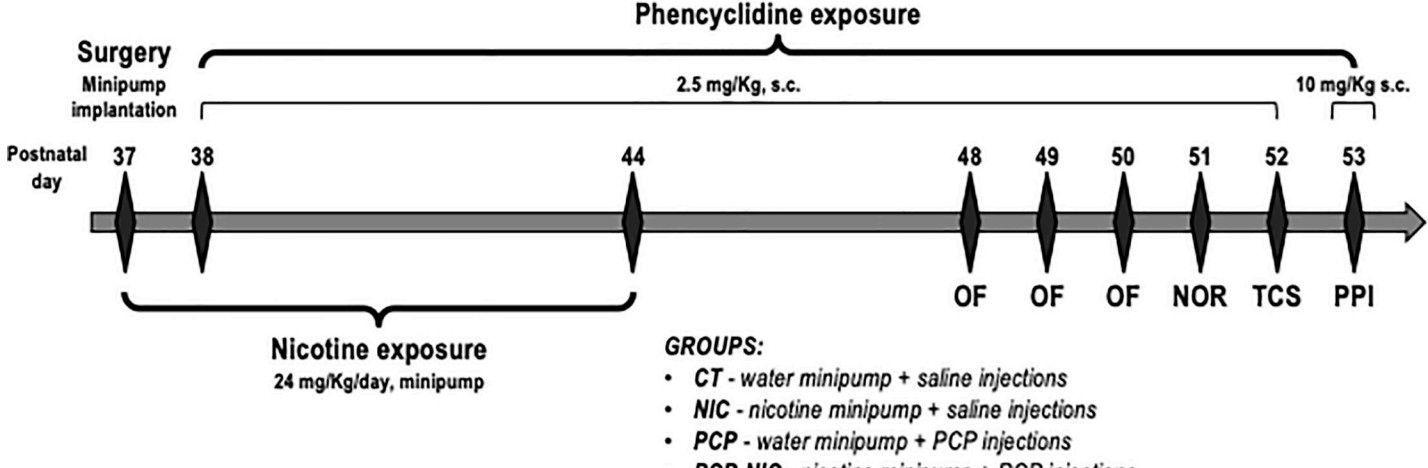

**Fig 1. Timeline of experimental design.** CT: Control group; NIC: Nicotine-exposed group; NOR: Novel Object Recognition test; OF: Open Field test; PCP: Phencyclidine-treated group; PCP-NIC: Phencyclidine-treated and nicotine-exposed group; PN: Postnatal day; PPI: Prepulse inhibition test; TCS: Three-Chamber Sociability test.

2–5 at 21–22°C on a 12-h light/dark cycle (lights on at 1:00 a.m.). Access to food and filtered water was *ad libitum*. Animals were derived from a C57BL/6 colony maintained at the Universidade Federal Fluminense (Niteroi, Brazil) for over 60 generations. Considering NIH recommendations [49] and previous evidence of sex differences in the severity of symptoms in SCHZ patients [50], both males and females were used.

From postnatal (PN) 37 to PN44, mice were exposed to nicotine. Briefly, at PN37, mice were anesthetized with xylazine (20 mg/Kg, i.p) and ketamine (100 mg/Kg, i.p), a small area on the back was shaved, and an incision was made to permit s.c. insertion of an osmotic minipump (model 1007d, diffusion rate: 0.5 μL/h, Alzet, Cupertino, CA, U.S.A.). Minipumps were prepared on the day preceding implantation with nicotine free base diluted in milli-Q water (pH adjusted to 6.0 with 5M NaOH) so as to deliver an initial dose rate of 24 mg/Kg/day of nicotine. Incisions were closed after minipump insertion and, both right after closure and on the following day, mice were given s.c. injections of flunixin (2.5 mg/Kg, s.c. 2.5 mL/Kg body mass) and enrofloxacin (2.5 mg/Kg, s.c. 2.5 mL/Kg body mass) for pain management and infection prevention. The period of exposure to nicotine intended to parallel human exposure during adolescence, a key period for initiation of tobacco and ENDS consumption [17,18,51]. The dose rate used in the current study was chosen based on a previous study that, by exposing mice to the same dose, produced nicotine serum levels [39] that are within the range of those found in smokers [52,53]. The dose choice also took into consideration evidence that adolescent rodents require higher nicotine doses to produce similar cotinine levels when compared to adults [54,55]. Control mice were implanted with minipumps containing milli-Q water. Animals were allowed to recover from surgery in their home cages. Mice were monitored twice a day for two days for fur appearance and behavior. There was no evidence of suffering and there were no deaths during surgery and postoperative period.

The antagonism of NMDAR produces behavioral and neurochemical abnormalities associated with SCHZ [42]. From PN38 to PN52, the animals were treated with daily injections of phencyclidine, an NMDA receptor antagonist used to model SCHZ, at a dose of 2.5 mg/Kg (s.c., dissolved in saline and administered at a volume of 2.5 mL/Kg body mass) [56,57]. On the last day of phencyclidine treatment (PN53), the dose used was 10 mg/Kg (s.c., 2.5 mL/Kg body mass) [58]. The period of phencyclidine treatment was chosen based on evidence that the

prodromal stage of the disorder begins early, still during adolescence [14–16] and that, in most cases, the first psychotic event in SCHZ occurs between late adolescence and early adulthood [16,59]. Control mice were injected with a saline solution (NaCl 0.9%). Accordingly, mice were distributed into four experimental groups: CT (control group), PCP (phencyclidine-treated group), NIC (nicotine-exposed group) and PCP-NIC (phencyclidine-treated and nicotine-exposed group). One hundred and ten mice were used. Between 11 and 16 animals were used in each experimental group and sex.

## 2.3. Behavioral tests (Fig 1)

The behavioral tests were conducted from PN48 to PN53, i.e., after the end of nicotine exposure, but still during the period of phencyclidine treatment. The 4-day interval between the end of nicotine exposure and the beginning of behavioral tests aimed to assess lasting effects of nicotine without the presence of confounding factors associated with acute withdrawal signs, which peak 24 h and are no longer evident 48 h after the end of exposure in mice [60–64]. Phencyclidine or saline injections occurred 10 min before each test. Body mass was measured every day during the treatment period.

All mice were submitted to the four behavioral tests described below (Fig 1). From PN48 to PN50, locomotor activity was assessed in the open field (OF). The repeated testing sessions further allowed the investigation of locomotor sensitization, a progressive increase in locomotion in response to repeated administration of a psychostimulant [65]. The OF was also used to assess anxiety-like behavior. However, considering evidence that retesting may alter both behavioral baselines and the nature of emotional responses in animal models of anxiety [66–68], anxiety-like behavior was assessed at PN48. On PN51, mice were submitted to the two sessions of the novel object recognition (NOR) test to assess memory/learning. On PN52, the three-chamber sociability (TCS) test was performed and, on PN53, the prepulse inhibition (PPI) test investigated the integrity of sensorimotor gating. The order of the behavioral tests was based on the stress level mice are submitted during each test. The OF is considered a test that causes little stress [69]. The NOR test followed the OF because the same arena was used to both behavioral tests, and the periods of OF testing were used as habituation to the NOR. The PPI was the last due to its stressful nature (the animals are repeatedly submitted to sudden, intense auditory stimuli and restrained for the whole duration of the test) and to its stability (it does not exhibit habituation or extinction over multiple trials) [44]. All tests were performed during the same period of the circadian cycle of the mice (the light phase of our animal facility), in a sound attenuated room and with lights on (40 W fluorescent light bulb, 3 m high). The animals were allowed to acclimatize to the test room for at least 10 min before each test. The test apparatuses were cleaned with paper towels soaked in 35% ethanol and dried before each test. After the behavioral tests, the animals were anesthetized with xylazine (20 mg/Kg, i.p) and ketamine (100 mg/Kg, i.p) and decapitated.

**2.3.1. Open field test.** The behavioral phenotype of hyperlocomotor activity reflects increased dopaminergic activity in the mesolimbic system, which is associated with the positive symptomology of SCHZ [70]. The OF arena (Insight, SP, Brazil) consists of a transparent acrylic box (46 cm length × 46 cm width × 43 cm height) that is equipped with 2 arrays of 16 infrared beams each, positioned at 1.5 cm above the floor to measure horizontal spontaneous locomotor activity. Interruptions of photocell beams are detected by a computer system and the location of the animal is calculated by the software with a 0.1-s resolution. Mice were individually placed in the center of the arena and were allowed to explore it for 20 min. Spontaneous locomotor activity (Ambulation) was determined on the basis of the traversed distance. Anxiety disorders are highly prevalent in SCHZ [71] and anxiety comorbidity negatively

affects SCHZ patients functioning and quality of life [72]. In this sense, considering that, in rodents, measures of central exploration in the OF are often regarded as anxiety-related indices [73], the time spent in the center (Time Cen) was used as a measure of anxiety-like behavior. Increased Time Cen corresponds to decreased anxiety-like behavior and vice versa [73,74].

**2.3.2. Novel object recognition test.**   Recognition memory was explored using the NOR test. This test is analogous in some ways to human declarative (episodic) memory, one of the abnormal cognitive domains in SCHZ [75]. It relies on the rodent's tendency to explore novel objects over familiar ones, reflecting its memory of the latter. The NOR consists of 2 sessions. For the training session, mice were individually placed in the center of the arena and were allowed to explore two identical objects (A and B) on opposite sides (at a distance of 10 cm from the walls) for 5 min. Animals were then returned to their home cages. For the test session, 90 min later, object B was replaced with object C, which was novel to the animals and different from either object A or B. Mice were allowed to explore objects A and C for a period of 5 min. The time spent in exploring each of the objects was recorded in both sessions. A preference index (PI) was calculated as the ratio between time spent exploring the novel object and time spent exploring both objects.

**2.3.3. Three-chamber sociability test.**   Deficits in social interaction in animal models of SCHZ represent aspects of the negative symptomology of SCHZ [70,76]. The apparatus [77] is a rectangular clear plexiglass box divided into three chambers. The chambers are 26 cm length × 12 cm width × 50 cm height, with 10-cm wide openings into the two end chambers. The chambers are isolated with two dividing plexiglass walls. In each side chamber, there is a cylinder made of grid bars (1-cm distance between adjacent grid bars). The experimental mouse was placed in the middle chamber and allowed to explore the entire apparatus for 10 min. After that, the animal was returned to its home cage and an "unfamiliar" mouse was placed in one side chamber (social side) inside the cylinder. The other cylinder, on the other side chamber (non-social side) was kept empty. The "unfamiliar" mouse was of the same sex and age of the experimental animal. The experimental mouse was then placed back in the middle chamber and allowed to explore the entire apparatus for 10 min. The percent time on social side [%Time = (social side/(social side + middle chamber + non-social side)) * 100] and the time to first entry into the social side (Latency) were used as measures of social interaction. Entries (number of entries in social side + non-social side) was used as a measure of activity. The mouse was considered to be in a given chamber when its head and 2 front paws were inside it.

**2.3.4. Prepulse inhibition test.**   Deficits in PPI indicate disrupted sensorimotor gating. In healthy individuals, a small acoustic stimulus (prepulse) given before a strong startle-eliciting stimulus (startle pulse) attenuates the response to the startle-eliciting stimulus so that an individual can allocate attentional resources to salient environmental stimuli [44]. This ability is impaired in patients with SCHZ in whom PPI deficits significantly correlate with cognitive deficits [44,78]. Startle responses were measured using the Panlab Startle and Fear combined system (Harvard Apparatus, Massachusetts, U.S.A.). Each mouse was placed in the startle apparatus with a 65-decibel (dB) background white noise for a 5-min acclimatization period. Following this period, 10 startle pulse trials (Sp, 120 dB, 40 ms) were presented. Next, 50 semi-randomized trials consisting of no pulse, a Sp (120 dB, 40 ms) or a prepulse (Pp either 70 dB, 75 dB, or 80 dB, 40 ms) were presented 100 ms before the Sp. These were followed by 10 Sp trials (120 dB, 40 ms). For all trials, a fixed 20 s interval was used. The startle responses were measured every 1 ms during the 100 ms period that followed the presentation of the startle stimulus. The amplitudes of the startle responses were averaged separately for each trial. The degree of prepulse inhibition is shown as the percentage of inhibition (% Inhibition), according to the following formula: % Inhibition = 100 −(PpSp × 100)/Sp, where PpSp is the mean

response amplitude for prepulse-plus-startle pulse trials and Sp is the mean response amplitude for the startle pulse alone trials. Only the startle responses to the semi-randomized trials were used in this formula. Based on previous studies, we chose a PCP dose of 10 mg/Kg to be administered before the PPI test [58].

## 2.4. Statistical analysis

To reduce the likelihood of type 1 statistical errors that might result from repeated testing, results on each variable were evaluated first by global univariate analysis of variance (uANOVAs) or repeated measures analyses of variance (rANOVAs). rANOVAs were performed for the analyses of body mass, OF (Ambulation), NOR (PI) as well as PPI (% Inhibition) variables. uANOVAs were performed for Time Cn in the OF and for the TCS variables (%Time, Latency and Entries). Within-subject factors were: Day for body mass and OF data; Session for NOR data, and dB level for PPI data. Phencyclidine (treated: PCP and PCP-NIC groups; non-treated: CT and NIC groups), Nicotine (exposed: NIC and PCP-NIC groups; non-exposed: CT and PCP groups) and Sex were the between-subject factors for all variables. Lower-order uANOVAs, Fisher Protected Least Significant Difference (FPLSD) tests or paired t-tests were used post hoc where appropriate. Figures were segmented by sex specifically when significant Sex interactions were observed.

All data were analyzed using the IBM SPSS Statistics for Windows, Version 21.0 (IBM Corp, Armonk, N.Y., U.S.A.) and compiled as means and standard errors. Significance was assumed at the level of $p < 0.05$ for main effects; however, for interactions at $p < 0.1$, we also examined whether lower-order main effects were detectable after subdivision of the interactive variables [79]. The criterion for interaction terms was not used to assign significance to the effects but rather to identify interactive factors requiring subdivision for lower-order tests of main effects of Phencyclidine and/or Nicotine, the factors of chief interest [79].

# 3. Results

## 3.1. Body mass

Neither phencyclidine treatment nor nicotine history affected body mass (Table 1).

## 3.2. Open field fest (Fig 2)

The rANOVA revealed that locomotor activity varied as a function of phencyclidine treatment, nicotine history and test day (Ambulation: Phencyclidine–$F_{1,102} = 96$, $p < 0.0001$; Day × Nicotine–$F_{2,204} = 3.7$, $p = 0.025$; Day × Phencyclidine × Nicotine–$F_{2,204} = 2.9$, $p = 0.056$). Accordingly, each day was analyzed separately. The uANOVA on day 1 identified a hyperlocomotor effect of phencyclidine treatment (Phencyclidine–$F_{1,102} = 53.8$, $p < 0.0001$): phencyclidine-treated mice (PCP and PCP-NIC) were more active than non-treated ones (CT and NIC) (Fig 2A). On days 2 (Phencyclidine–$F_{1,102} = 76.4$, $p < 0.0001$; Phencyclidine × Nicotine–$F_{1,102} = 4.1$, $p = 0.046$) and 3 (Phencyclidine–$F_{1,102} = 81.1$, $p < 0.0001$; Phencyclidine × Nicotine–$F_{1,102} = 5.5$, $p = 0.021$) even though the hyperlocomotor effect of phencyclidine treatment was still identified (days 2 and 3: PCP > CT, PCP > NIC, PCP-NIC > CT, PCP-NIC > NIC, $p < 0.0001$ for all pairwise comparisons, FPLSD), nicotine history mitigated this effect (day 2: PCP > PCP-NIC, $p = 0.03$; day 3: PCP > PCP-NIC, $p = 0.008$, FPLSD) (Fig 2A). Interestingly, paired t-tests for each experimental group further identified a progressive increase in locomotion from day 1 to day 2 ($p = 0.039$) and from day 1 to day 3 ($p = 0.015$), in PCP mice (Fig 2A). This increase is consistent with a phencyclidine-evoked locomotor sensitization. A similar increase was not identified in PCP-NIC mice, which

**Table 1. Body mass gain throughout the experimental period.**

| | Body mass (g) | | | | | | | | | | | | | | | | |
|---|---|---|---|---|---|---|---|---|---|---|---|---|---|---|---|---|---|
| | *Females* | | | | | | | | | | | | | | | | |
| | PN37 | PN38 | PN39 | PN40 | PN41 | PN42 | PN43 | PN44 | PN45 | PN46 | PN47 | PN48 | PN49 | PN50 | PN51 | PN52 | PN53 |
| CT | 16.2 ±0.3 | 16.2 ±0.2 | 16.4 ±0.2 | 16.5 ±0.2 | 16.7 ±0.3 | 16.9 ±0.3 | 16.9 ±0.2 | 17.0 ±0.3 | 17.1 ±0.3 | 17.2 ±0.3 | 17.2 ±0.3 | 17.3 ±0.3 | 17.5 ±0.3 | 17.6 ±0.3 | 17.6 ±0.3 | 17.7 ±0.3 | 17.7 ±0.2 |
| NIC | 16.5 ±0.3 | 16.7 ±0.4 | 17.0 ±0.5 | 17.2 ±0.6 | 16.9 ±0.4 | 17.3 ±0.4 | 17.2 ±0.4 | 17.4 ±0.4 | 17.2 ±0.4 | 17.4 ±0.4 | 17.5 ±0.4 | 17.6 ±0.4 | 18.1 ±0.4 | 18.0 ±0.4 | 18.0 ±0.4 | 18.2 ±0.4 | 18.2 ±0.4 |
| PCP | 16.2 ±0.4 | 16.2 ±0.3 | 16.5 ±0.3 | 16.4 ±0.3 | 16.5 ±0.3 | 16.6 ±0.3 | 16.6 ±0.3 | 16.7 ±0.3 | 16.8 ±0.3 | 16.8 ±0.3 | 16.9 ±0.3 | 17.0 ±0.3 | 17.3 ±0.4 | 17.3 ±0.3 | 17.3 ±0.3 | 17.4 ±0.3 | 17.4 ±0.3 |
| PCP-NIC | 16.7 ±0.3 | 16.5 ±0.3 | 16.5 ±0.3 | 16.9 ±0.3 | 17.1 ±0.3 | 17.3 ±0.3 | 17.4 ±0.2 | 17.5 ±0.3 | 17.6 ±0.3 | 17.7 ±0.3 | 17.8 ±0.3 | 18.0 ±0.3 | 18.1 ±0.3 | 18.2 ±0.3 | 18.3 ±0.3 | 18.4 ±0.3 | 18.4 ±0.3 |
| | *Males* | | | | | | | | | | | | | | | | |
| | PN37 | PN38 | PN39 | PN40 | PN41 | PN42 | PN43 | PN44 | PN45 | PN46 | PN47 | PN48 | PN49 | PN50 | PN51 | PN52 | PN53 |
| CT | 18.4 ±0.4 | 18.7 ±0.4 | 19.0 ±0.4 | 19.2 ±0.4 | 18.9 ±0.7 | 19.6 ±0.4 | 19.9 ±0.3 | 20.0 ±0.4 | 20.1 ±0.3 | 20.2 ±0.4 | 20.5 ±0.4 | 20.7 ±0.4 | 20.8 ±0.3 | 20.9 ±0.3 | 21.0 ±0.4 | 21.1 ±0.3 | 21.1 ±0.4 |
| NIC | 19.1 ±0.4 | 19.3 ±0.4 | 19.4 ±0.4 | 19.6 ±0.3 | 19.8 ±0.3 | 20.1 ±0.3 | 20.4 ±0.3 | 20.5 ±0.3 | 20.6 ±0.3 | 20.9 ±0.3 | 21.0 ±0.3 | 21.3 ±0.4 | 21.5 ±0.3 | 21.6 ±0.3 | 21.7 ±0.3 | 21.6 ±0.3 | 21.7 ±0.3 |
| PCP | 19.0 ±0.3 | 19.2 ±0.3 | 19.0 ±0.3 | 19.3 ±0.3 | 19.7 ±0.3 | 20.1 ±0.2 | 20.0 ±0.3 | 20.1 ±0.2 | 20.4 ±0.2 | 20.4 ±0.2 | 20.5 ±0.2 | 20.7 ±0.2 | 20.7 ±0.2 | 20.9 ±0.2 | 21.0 ±0.2 | 20.9 ±0.2 | 21.1 ±0.2 |
| PCP-NIC | 18.6 ±0.5 | 18.6 ±0.4 | 18.9 ±0.4 | 19.1 ±0.3 | 19.3 ±0.4 | 19.6 ±0.3 | 19.7 ±0.3 | 20.0 ±0.3 | 19.9 ±0.4 | 20.2 ±0.3 | 20.3 ±0.3 | 20.4 ±0.4 | 20.6 ±0.3 | 20.7 ±0.4 | 20.8 ±0.4 | 21.0 ±0.3 | 21.1 ±0.3 |

CT: Control group (♀ = 12, ♂ = 16); NIC: Nicotine-exposed group (♀ = 14, ♂ = 14); PCP: Phencyclidine-treated group (♀ = 14, ♂ = 14); PCP-NIC: Phencyclidine-treated and nicotine-exposed group (♀ = 12, ♂ = 14). PN: Postnatal day. Values are means ± SEM.

further demonstrates that nicotine history reduced the impact of phencyclidine on locomotion.

Time Cn on day 1, was used do assess anxiety-like behavior. The uANOVA (Time Cn: Phencyclidine × Nicotine–$F_{1,102}$ = 3.1, $p$ = 0.083) identified reductions in the time spent in the center of the OF arena in response to phencyclidine treatment (CT > PCP, $p$ = 0.042, FPLSD), nicotine history (CT > NIC, $p$ = 0.008, FPLSD) and the combined insult (CT > PCP-NIC, $p$ = 0.041, FPLSD) (Fig 2B), which indicates anxiogenic effects.

### 3.3. Novel object recognition (Fig 3)

The rANOVA demonstrated that both phencyclidine exposure and nicotine history affected the time the animals explored the objects, expressed as the PI. However, the effects varied as a function of the session (Session × Phencyclidine–$F_{1,101}$ = 8.0, $p$ = 0.006; Session × Phencyclidine × Nicotine–$F_{1,101}$ = 3.9, $p$ = 0.050) and sex (Session × Phencyclidine × Nicotine × Sex–$F_{1,101}$ = 4.9, $p$ = 0.028). Separate lower-order uANOVAs on each session failed to show significant differences in the training session (Fig 3). In contrast, in the test session, the uANOVA confirmed sex-selective effects (Phencyclidine–$F_{1,101}$ = 7.0, $p$ = 0.009; Phencyclidine × Nicotine × Sex–$F_{1,101}$ = 3.4, $p$ = 0.070).

Separate analysis for males and females revealed that, in females (Test session: Phencyclidine × Nicotine–$F_{1,48}$ = 3.9, $p$ = 0.054), a higher preference for the new object was evident in NIC mice when compared to the CT (Test session: NIC > CT, $p$ = 0.046, FPLSD) and PCP-NIC (Test session: NIC > PCP-NIC, $p$ = 0.02, FPLSD) ones. Paired t-tests for each experimental group further confirmed the impact of nicotine (Fig 3A): Only NIC females increased the preference for the new object when the training and the test sessions were compared ($p$ < 0.001). As for males, a smaller preference for the new object was evident in

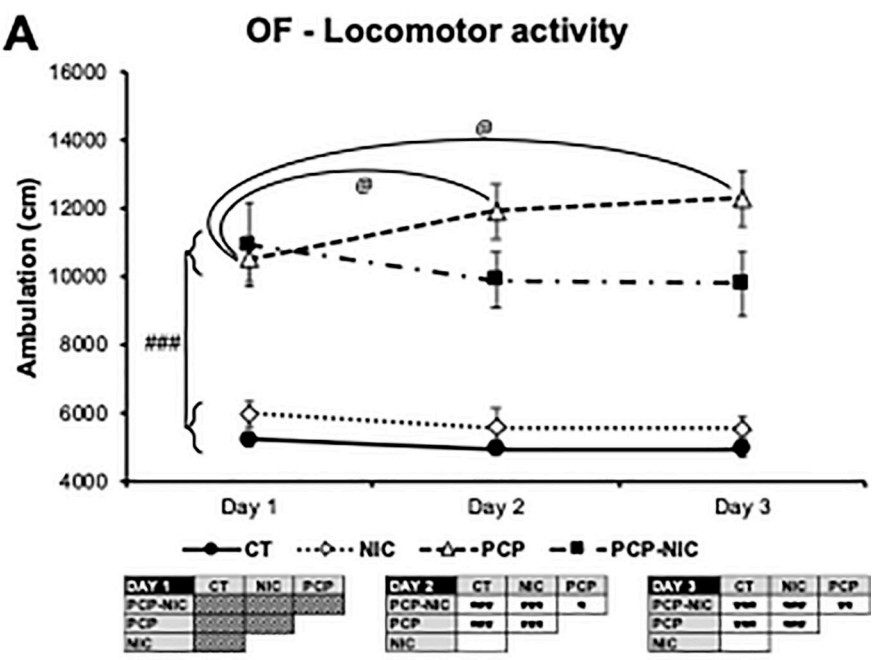

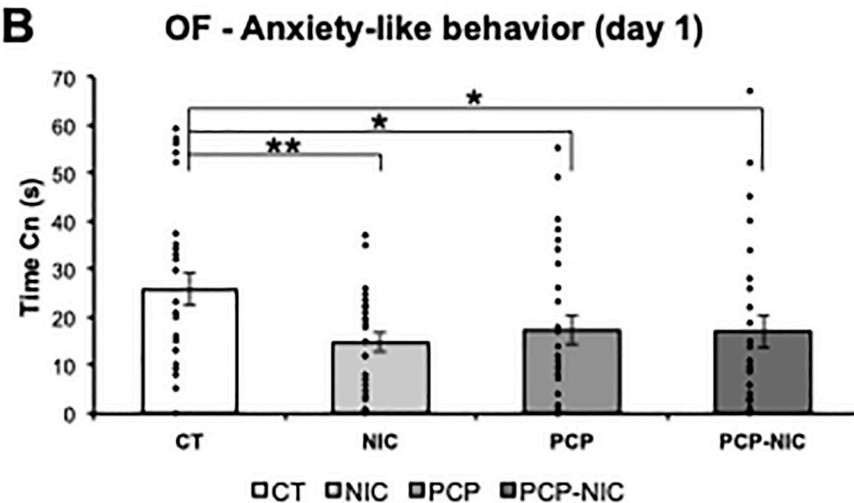

**Fig 2.** Effects of phencyclidine treatment, nicotine history and the combined insults on adolescent mice, as assessed in the open field arena: (A) locomotor activity and (B) anxiety-like behavior. CT: Control group ($\female$ = 12, $\male$ = 16); NIC: Nicotine-exposed group ($\female$ = 14, $\male$ = 14); PCP: Phencyclidine-treated group ($\female$ = 14, $\male$ = 14); PCP-NIC: Phencyclidine-treated and nicotine-exposed group ($\female$ = 12, $\male$ = 14). Values are means ± SEM. *$p < 0.05$, **$p < 0.01$, ***$p < 0.001$. @$p < 0.05$, Day 1 *vs*. Day 2 or Day 1 *vs*. Day 3 within each experimental group. ###$p < 0.001$, phencyclidine-treated (PCP and PCP-NIC) *vs*. phencyclidine non-treated (CT and NIC) mice.

phencyclidine-treated mice (PCP and PCP-NIC) when compared to non-treated ones (CT and NIC) (Test session: Phencyclidine–$F_{1,53} = 6.0$, $p = 0.017$). The negative impact of phency-clidine on memory was further confirmed by paired t-tests: While both CT ($p = 0.011$) and NIC ($p < 0.001$) males increased the preference for the new object when the training and the test sessions were compared, there were no differences between sessions for PCP and PCP-NIC ones (Fig 3B).

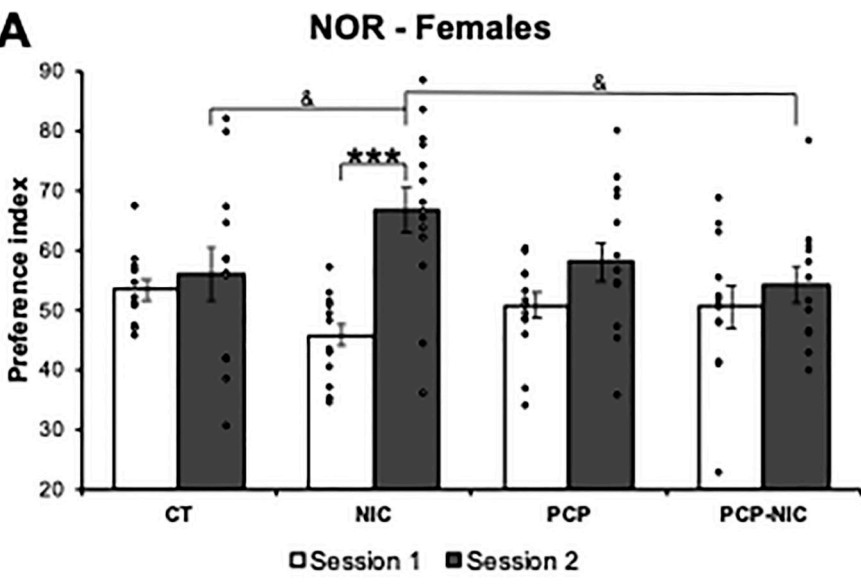

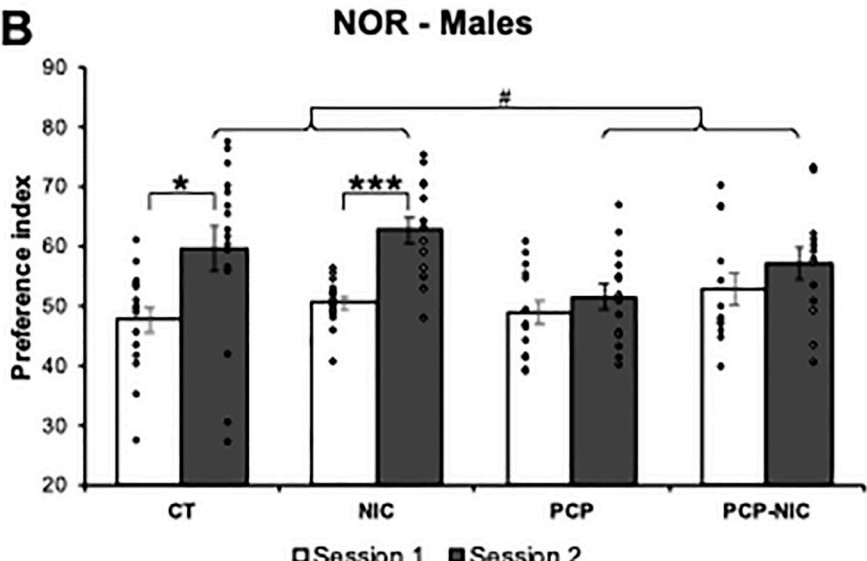

**Fig 3.** Effects of phencyclidine treatment, nicotine history and the combined insults on the preference index of adolescent (A) female and (B) male mice, as assessed in the novel object recognition test. CT: Control group (♀ = 12, ♂ = 16); NIC, Nicotine-exposed group (♀ = 14, ♂ = 14); PCP: Phencyclidine-treated group (♀ = 14, ♂ = 14); PCP-NIC: Phencyclidine-treated and nicotine-exposed group (♀ = 12, ♂ = 13). S1: Session 1; S2: Session 2. Values are means ± SEM. $^{*}p < 0.05$, $^{***}p < 0.001$, S1 *vs.* S2 within each experimental group. $^{\#}p < 0.05$, phencyclidine-treated (PCP and PCP-NIC) *vs.* phencyclidine non-treated (CT and NIC) mice in Session 2. $^{\&}p < 0.05$, NIC *vs.* CT and PCP-NIC mice in Session 2.

### 3.4. Three-chamber sociability test (Fig 4)

The time to first entry into the social side of the TCS apparatus was affected by interactive effects between phencyclidine and nicotine (Latency: Nicotine–$F_{1,90}$ = 6.5, $p$ = 0.013; Phencyclidine × Nicotine–$F_{1,90}$ = 6.1, $p$ = 0.015). Phencyclidine treatment increased the Latency (PCP > CT, $p$ = 0.003; PCP > NIC, $p$ = 0.002, FPLSD), while nicotine history mitigated this effect (PCP > PCP-NIC, $p$ = 0.001, FPLSD) (Fig 4A). The %Time on social side failed to show significant effects (Fig 4B). Similar to the increased locomotor activity identified

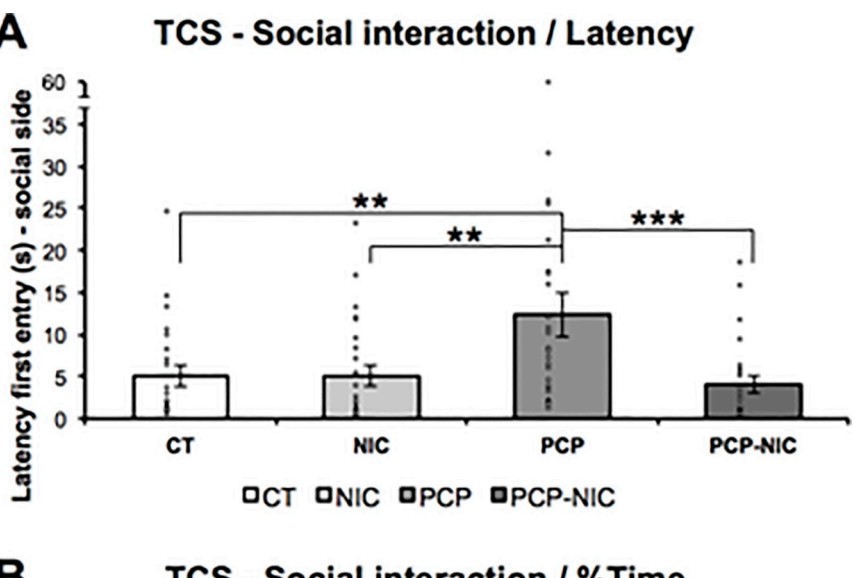

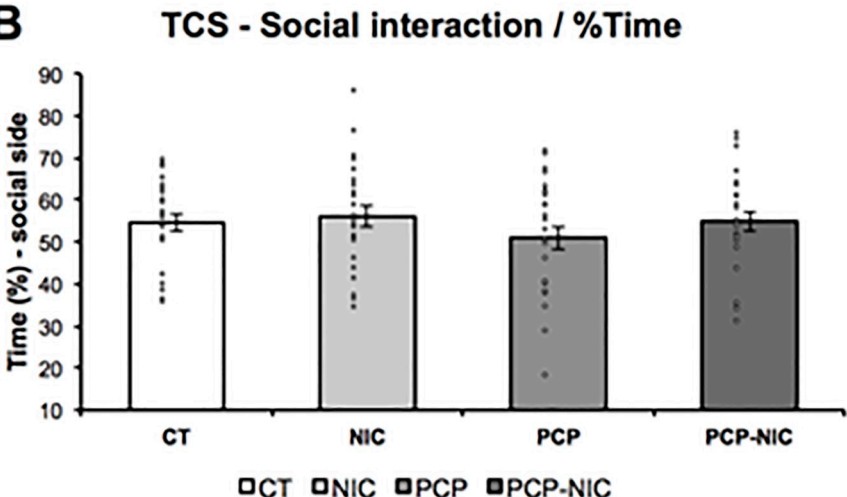

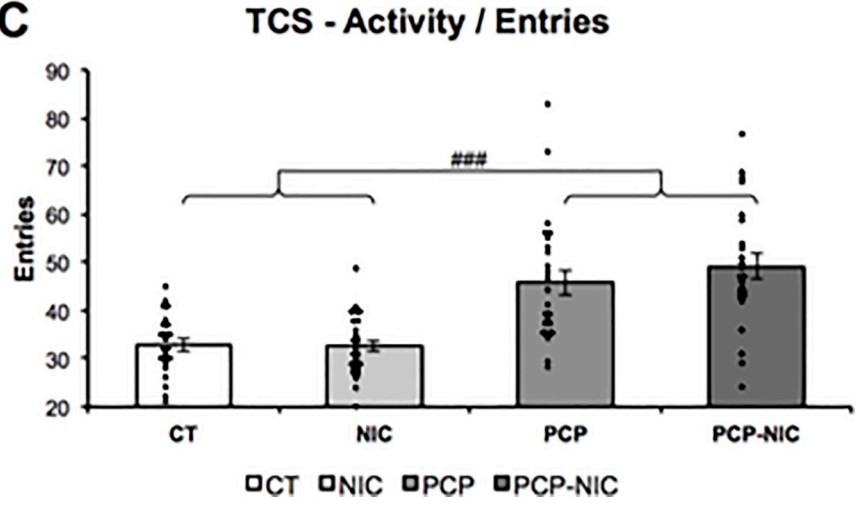

**Fig 4.** Effects of phencyclidine treatment, nicotine history and the combined insults on adolescent mice's (A) time to first entry into social side, (B) percent time on social side and (C) activity, as assessed in the three-chamber sociability test. CT: Control group (♀ = 11, ♂ = 12); NIC: Nicotine-exposed group (♀ = 12, ♂ = 14); PCP: Phencyclidine-treated

group ($♀$ = 11, $♂$ = 14); PCP-NIC: Phencyclidine-treated and nicotine-exposed group ($♀$ = 12, $♂$ = 12). Values are means ± SEM. $^{**}p < 0.01$, PCP *vs*. CT and NIC mice. $^{***}p < 0.001$, PCP *vs*. PCP-NIC mice. $^{###}p < 0.001$, phencyclidine-treated (PCP and PCP-NIC) *vs*. phencyclidine non-treated (CT and NIC) mice.

on the first day of test in the OF, in the TCS, an increase in activity, calculated as the number of entries in the social side plus the non-social side (Entries: Phencyclidine–$F_{1,90}$ = 48.5, $p < 0.001$) was identified in phencyclidine-treated mice (PCP and PCP-NIC) when compared to non-treated ones (CT and NIC) (Fig 4C).

### 3.5. Prepulse inhibition test (Fig 5)

The rANOVA revealed a deficient PPI in response to phencyclidine treatment, however, this response varied as a function of the Pp dB level (dB level × Phencyclidine–$F_{2,204}$ = 6.6, $p = 0.002$). The lower-order uANOVAs on each Pp dB level showed a reduced PPI in phencyclidine-treated mice (PCP and PCP-NIC) when compared to non-treated ones (CT and NIC) when the higher Pp level (80 dB) preceded the Sp (Phencyclidine–$F_{1,102}$ = 8.2, $p = 0.005$).

## 4. Discussion

The epidemiological association between SCHZ and nicotine consumption is already evident during adolescence [13], a period of brain maturation during which there is increased susceptibility to nicotine [20] and that also represents a transitional period to SCHZ, when symptoms first appear and progressively aggravate [14–16]. The early association between nicotine consumption and SCHZ raises up the possibility that nicotine interference in the developing brain plays a central role in the development of SCHZ. Despite that, studies that investigate the interactions between SCHZ and nicotine classically use adult subjects. To the best of our knowledge, this is the first study that investigated whether a short period of nicotine exposure during adolescence changes the course of behaviors that model the distinct dimensions of SCHZ. Nicotine history reduced the magnitude of phencyclidine-evoked hyperlocomotion and impeded

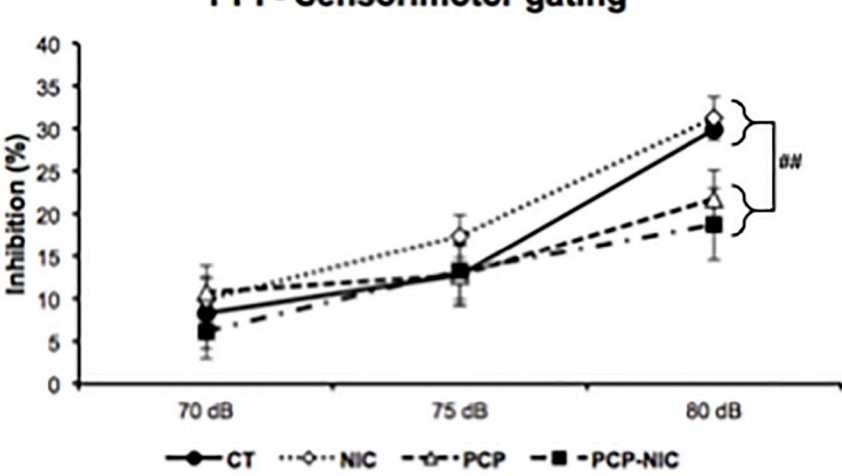

**Fig 5. Effects of phencyclidine treatment, nicotine history and the combined insults, on prepulse inhibition of adolescent mice.** CT: Control group ($♀$ = 12, $♂$ = 16); NIC: Nicotine-exposed group ($♀$ = 14, $♂$ = 14); PCP: Phencyclidine-treated group ($♀$ = 14, $♂$ = 13); PCP-NIC: Phencyclidine-treated and nicotine-exposed group ($♀$ = 13, $♂$ = 14). Values are means ± SEM. $^{##}p < 0.01$, phencyclidine-treated (PCP and PCP-NIC) *vs*. phencyclidine non-treated (CT and NIC) mice using the 80 dB prepulse.

the development of locomotor sensitization. It also reversed the deficient sociability elicited by phencyclidine. Distinctively, phencyclidine-evoked memory and sensorimotor gating deficits were neither improved nor worsened by prior nicotine exposure. These results show that SCHZ-like symptoms evoked by a phencyclidine-induced mice model are either ameliorated or unaffected by adolescent nicotine history. In the ensuing paragraphs, we elaborate these essential conclusions.

Psychotic episodes have been associated with increased dopaminergic activity in the mesolimbic system [26]. In SCHZ, the hyperfunctioning of this system may be due to an imbalance in mesolimbic signaling through VTA dopaminergic neurons and their output D1 and D2 receptor substrates in the nucleus accumbens, which results in D2 receptors overactivation [26]. In turn, in rats treated with nicotine during adolescence but not at adulthood, a marked decrease in the expression of accumbal D1 receptors was evident following exposure, which lasted until young adulthood [80]. Considering that the coordinated activation of D1 and D2 receptors is necessary for proper reward-guided behaviors to take place [81], this D1 receptors downregulation may counterbalance the D2 overactivation that is characteristic of SCHZ and, as a result, contribute both to the reduced locomotor activity in adolescent PCP-NIC mice when compared to PCP ones and to hampering the development of phencyclidine-evoked locomotor sensitization in PCP-NIC mice. Beneficial effects of nicotine might also be mediated by its capability to upregulate α4β2 nAChRs expressed in GABAergic neurons, as this effect could enhance the activity of inhibitory circuits and reduce VTA dopaminergic activity [82]. Importantly, adolescent rodents exposed to nicotine show robust and persistent upregulation [34,35], which corroborates the idea of a unique effect of adolescent nicotine on SCHZ symptomatology. Considering that the behavioral phenotype of hyperlocomotor activity is associated with the positive symptoms of SCHZ [70] and that the progressive increase of activity that characterizes locomotor sensitization is a well-known model of behavioral plasticity with features in common with the development of psychosis [65], our data provide evidence that nicotine history during adolescence improves positive symptoms of SCHZ.

Despite the potential mechanisms described above, nicotine history did not fully reverse the hyperlocomotor effect of phencyclidine, which suggests that other factors might hamper this beneficial effect. In this regard, nicotine binding to α7 nAChRs was shown to strengthen glutamatergic synapses in VTA dopaminergic neurons [20]. This is a long-term effect [83,84] that may lessen the impact of the decreased expression of accumbal D1 receptors and the enhanced activity of inhibitory circuits.

The mitigation of phencyclidine-evoked hyperlocomotion mediated by nicotine history was evident on the second and third days of the OF test but not on the first day. This chronological difference may indicate that the stress associated with the animals' exposure to a novel situation interfered with the outcome. The identification of an anxiogenic profile in the first OF day corroborates this possibility. In addition, since both in the first OF day and in the TCS there was no prior habituation to the test apparatuses, high stress levels are expected. In accordance with this assumption, similar activity levels in PCP-NIC mice when compared to PCP ones were identified in both situations. Glucocorticoids modulation of extracellular dopamine levels in the mesocorticolimbic system has been demonstrated [85,86], which provides a mechanistic explanation for the interference of the hypothalamic–pituitary–adrenal axis in the dopaminergic tone of the mesocorticolimbic system and suggests that stressful situations hamper beneficial effects of nicotine on the positive symptomatology of SCHZ.

Besides its established use to measure locomotor activity, the OF test is also useful to index emotional aspects of anxiety-like behavior in rodents [73]. Even though mood disorders are not classically listed as SCHZ symptoms, roughly 40% of SCHZ patients express anxiety disorders at syndromic level [71], and anxiety is related to functional impairment and reduced

quality of life in this population [72]. In parallel, both clinical and pre-clinical evidence associate adolescent nicotine exposure with increased vulnerability to developing anxiety disorders in later life [87]. In accordance with these findings, here, we show high anxiety levels in the nicotine-exposed and phencyclidine-treated groups when compared to the control one, which indicates that both insults act independently in leading to the same outcome. As for the concurrent insults, they also resulted in high anxiety levels, however these were comparable in magnitude to the outcome of each insult. This last result may be interpreted as a failure of adolescent nicotine history in worsening anxiety levels in mice treated with phencyclidine. It is conceivable that, if a similar outcome occurs in SCHZ patients, nicotine history during adolescence might not be a relevant contributor to SCHZ-associated anxiety symptoms.

The association of tobacco consumption with negative symptoms of SCHZ has received little attention although these symptoms have been associated with poor functional outcome and remain resistant to treatment with the available medication [88]. According to Galderisi and collaborators [88], asociality fits into the avolition-apathy domain of SCHZ negative symptoms and is related to deficits in motivation systems. In the current study, we failed to identify significant differences for the variable %Time, which is classically used to quantify social interaction in the TCS test. However, the time to first entry into the social side of the TCS was increased in PCP mice, which suggests decreased sociability. Even though nicotine history per se failed to affect this variable, it reversed the phencyclidine-evoked effect. Negative symptoms are a heterogeneous group of SCHZ symptoms for which the pathophysiological mechanisms that are involved are still under investigation, however, it has been suggested that these symptoms are associated with a reduced dopaminergic neurotransmission in the mesocortical dopaminergic system [26,88,89], which might be improved by the modulation of dopaminergic function evoked by drugs of abuse [90,91]. In this regard, recent evidence points to an important role of both α7 and α4β2 nAChRs in the beneficial effect of acute nicotine in social behavior impairment in adult mice exposed to phencyclidine [92].

Together with negative symptoms, cognitive deficits are poorly responsive to both typical and atypical antipsychotic drugs [93], being considered the most relevant factor leading to poor functional outcome in SCHZ [75]. Here, by using the NOR test, a non-rewarded paradigm widely accepted as a rodents' model of human declarative memory [75], we demonstrated that male but not female cognition was harmed by phencyclidine treatment. This finding is in accordance with previous evidence of more severe cognitive impairment in male SCHZ patients when compared to female ones [50] and with results derived from both neurodevelopmental [94,95] and genetic animal models of SCHZ [95] tested at young adulthood. As for nicotine, its impact was null, as there was neither improvement nor further deterioration in phencyclidine-evoked cognitive deficits. To our knowledge, this is the first study that investigates cognition in a phencyclidine-induced mice model of SCHZ and adolescent nicotine history. However, this pattern of results is in tight agreement with a study that used a SCHZ model produced by neonatal ventral hippocampal lesions. Berg and collaborators [36] showed deficits in contextual-working memory in adolescent mice with hippocampal lesions and, in these animals, nicotine history neither significantly worsened nor ameliorated SCHZ-like cognition deficits. Mice from this SCHZ model also exhibited downregulation of nAChRs in the frontal cortex, however the expected increase in the expression of nAChRs in response to nicotine history was equivalent in both control and lesioned mice [36]. These and our current findings agree with human data suggesting that exposure to nicotine through cigarette smoking is not associated with cognitive functioning in first-episode young adult SCHZ patients [96,97]. Future studies are needed to verify whether there is abnormal cholinergic signaling in our phencyclidine-induced adolescent mice model of SCHZ. It should also be noted that declarative memory is one of several cognitive domains disrupted in SCHZ [98]. The impact of

adolescent nicotine history on other cognitive domains affected by SCHZ warrants further investigation.

Similar levels of PPI deficits were identified in PCP and PCP-NIC groups. PPI of the startle response occurs in mammals, from humans to rodents, making it potentially useful for cross-species translational research. Its level reflects the functioning of sensorimotor gating mechanisms, which greatly impacts cognitive processes, particularly sensory-cognitive integration and proper motor response implementation [44]. Both α4β2 and α7 nAChRs are involved in the PPI phenomenon [99–101], which suggests that neurobiological changes in the early course of SCHZ are alterable by nicotine. In this regard, an elegant study by Cadenhead [102] identified larger PPI in young subjects at risk for psychosis that smoke when compared to non-smokers. In contrast, in smokers in the early stages of psychosis, PPI was reduced when compared to non-smokers. These findings corroborate the possibility that compensatory changes that take place during early stages of SCHZ are affected by nicotine, so that with the progression of the disease (from the "at risk" to the "early stage" of psychosis), the impact of nicotine also changes. Since PPI is considered a SCHZ vulnerability marker in the SCHZ spectrum [103,104], it is surprising that, in the current study, adolescent nicotine history neither improved nor worsened phencyclidine-evoked deficits in PPI. While there might be several explanations for this lack of effect, it is possible that, in our model, the short-term period of nicotine exposure was not sufficient to evoke lasting effects on PPI, an explanation corroborated by similar PPI levels in both NIC and CT groups. A second attractive possibility is that, by exposing mice to phencyclidine and nicotine during the critical window of adolescence, we were able to model the transition from increased to reduced PPI described by Cadenhead [102]. Future studies with either shorter or longer periods of phencyclidine exposure during adolescence may be able to model young subjects at risk and in the early stages of psychosis, as well as the neurobiological changes that take place in the early course of SCHZ and in response to nicotine.

Despite its face, construct and predictive validities [40,42,105,106], the PCP model has caveats that should be taken in consideration. Some SCHZ symptoms are of human nature and, as such, a challenge to be modeled [107]. In contrast to pharmacological models, SCHZ is chronic and episodic, with different symptoms predominating at different stages of the disease [106]. Finally, even though the PCP model evokes adaptation mechanisms that correlate with findings obtained in patients with SCHZ, it relies on a single neurotransmitter construct [107]. Limitations are inherent to animal models and reinforce the need of studying SCHZ using multiple models associated with varied but complimentary characteristics of SCHZ. Another limitation of the current study is that nicotine and phencyclidine serum levels were not assessed. Accordingly, despite the lack of evidence in the literature, we cannot rule out that pharmacokinetic interactions between these drugs have impacted our results.

The association between tobacco smoking and SCHZ has gained attention in recent years [108]. However, scant studies in prodrome patients and in animal models of SCHZ investigated this comorbidity during adolescence, which has the potential to identify mechanisms of nicotine interference in the development of SCHZ. This investigation is particularly relevant, since recognizing the period of adolescence as critical for nicotine addiction and SCHZ comorbidity creates the opportunity of early intervention, which could mitigate deleterious outcomes of these diseases [109]. Here, by using animal models of adolescent exposure, we demonstrated that nicotine history during adolescence is associated with remediation of positive and negative symptoms. The possibility that nicotine, restricted to this critical period of development, normalizes SCHZ-related functional imbalances may have a significant positive impact on SCHZ prognosis. Most importantly, the current findings open new possibilities for the investigation of therapeutic approaches to be used during adolescence. In this regard,

future studies in adolescents should focus on α7 ligands and allosteric modulators, which have a positive impact on cognition and other processes relevant to SCHZ [110,111]. Varenicline, an approved smoking cessation aid in adults, has low addictive potential when compared to nicotine [112] and its activity as a full agonist of α7 and a partial agonist for the α4β2 nAChR subtypes may also prove to be useful [113–116].

## Supporting information

**S1 Raw data. Raw data.** CT: Control group; NIC: Nicotine-exposed group; NOR: Novel object recognition test; OF: Open field test; PCP: Phencyclidine-treated group; PCP-NIC: Phencyclidine-treated and nicotine-exposed group; PPI: Prepulse inhibition test; TCS: Three-chamber sociability test.
(XLSX)

## Acknowledgments

The authors are thankful to Ulisses Risso for animal care.

## Author Contributions

**Conceptualization:** Ana Carolina Dutra-Tavares, Alex C. Manhães, Claudio C. Filgueiras, Anderson Ribeiro-Carvalho, Yael Abreu-Villaça.

**Data curation:** Yael Abreu-Villaça.

**Formal analysis:** Alex C. Manhães, Yael Abreu-Villaça.

**Funding acquisition:** Ana Carolina Dutra-Tavares, Yael Abreu-Villaça.

**Investigation:** Ana Carolina Dutra-Tavares, Keila A. Semeão, Julyana G. Maia, Luciana A. Couto.

**Methodology:** Ana Carolina Dutra-Tavares, Keila A. Semeão, Yael Abreu-Villaça.

**Project administration:** Ana Carolina Dutra-Tavares, Keila A. Semeão, Yael Abreu-Villaça.

**Resources:** Alex C. Manhães, Claudio C. Filgueiras, Anderson Ribeiro-Carvalho.

**Supervision:** Ana Carolina Dutra-Tavares, Yael Abreu-Villaça.

**Validation:** Keila A. Semeão, Julyana G. Maia, Luciana A. Couto, Claudio C. Filgueiras.

**Visualization:** Keila A. Semeão, Julyana G. Maia.

**Writing – original draft:** Yael Abreu-Villaça.

**Writing – review & editing:** Ana Carolina Dutra-Tavares, Alex C. Manhães, Keila A. Semeão, Julyana G. Maia, Luciana A. Couto, Claudio C. Filgueiras, Anderson Ribeiro-Carvalho.

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
