## [Decision Letter · Decision Letter 0]

13 Jul 2021

PONE-D-21-15988

Does nicotine exposure during adolescence modify the course of schizophrenia-like symptoms? Behavioral analysis in a phencyclidine-induced mice model

PLOS ONE

Dear Dr. Abreu-Villaça,

Thank you for submitting your manuscript to PLOS ONE. After careful consideration, we feel that it has merit but does not fully meet PLOS ONE’s publication criteria as it currently stands. Therefore, we invite you to submit a revised version of the manuscript that addresses the points raised during the review process.

We look forward to receiving your revised manuscript.

Kind regards,

Sergio D. Iñiguez, Ph.D.

Academic Editor

PLOS ONE

Journal Requirements:

Reviewers' comments:

Reviewer's Responses to Questions

**Comments to the Author**

1. Is the manuscript technically sound, and do the data support the conclusions?

Reviewer #1: Partly

Reviewer #2: Partly

Reviewer #3: Partly

2. Has the statistical analysis been performed appropriately and rigorously? 

Reviewer #1: No

Reviewer #2: Yes

Reviewer #3: No

3. Have the authors made all data underlying the findings in their manuscript fully available?

Reviewer #1: Yes

Reviewer #2: Yes

Reviewer #3: Yes

4. Is the manuscript presented in an intelligible fashion and written in standard English?

Reviewer #1: Yes

Reviewer #2: Yes

Reviewer #3: Yes

5. Review Comments to the Author

Reviewer #1: Dutra-Tavares et al. provide a series of behavioral studies examining the interaction between chronic nicotine exposure and phencyclidine-induced schizophrenia in mice. The authors report significant impairments in locomotor stimulant, negative affective, social and sensorimotor gating behavior induced by repeated phencyclidine treatment that are mitigated somewhat by concurrent nicotine exposure. The authors provide discussion on the possibility that nicotine (as a function of its agonist properties on nicotinic receptors) may have some therapeutic effects that warrant further investigation. While an interesting study, there are a few fundamental questions regarding the experimental design and statistical analysis. Moreover, there is some liberty taken at the explanation of what may be driving these behavioral differences, although the data presented here are largely devoid of neurobiological examinations. That being said, the findings are interesting and address novel aspects of the etiology of schizophrenic neuropathology during the adolescent period.

Major Considerations:

1. It is somewhat difficult to make a thorough conclusion of the study because the experimental design may require additional controls. For instance, in my reading of the study, it appears that mice either received nicotine or saline pumps, accompanied by phencyclidine or saline injections and only received these treatments throughout multiple behavioral tests (i.e., either phencyclidine or saline). Is it possible that an acute injection of PCP on test day could induce behavioral responses that would be important for understanding the basis of nicotine-phencyclidine interactions? If so, there appear to be several groups missing in the experimental design (e.g., nicotine and saline pumps + saline injections, given PCP on test days). Likewise, it would seem prudent to compare the effects of mice receiving PCP injections throughout behavioral testing with relevant saline controls (e.g., nicotine and saline pumps + PCP injections, given saline on test days).

2. The authors utilize a statistical approach that requires further explanation. The authors cite the use of both Fisher’s PLSD and relaxed criteria for qualifying interactions. There are several places in the Results section where the interactions are subthreshold (p=0.056, p.13; p=0.083, p.14; p=0.07 and 0.054, p.15), and still probed for significance changes without adjusting for multiple pairwise comparisons. How do the authors’ justify this approach? Planned comparisons, data transformations, outlier detection all seem to be better options for consideration. Careful detail/justification should be included in the Statistics paragraph of the Methods section.

3. It is not clear whether the study was intended to explore sex differences? There is hardly any rationale given and no hypothesis stated in the Introduction. Moreover, the approach employed for analyzing sex differences appears inconsistent (e.g., why only probe sex differences in novel object recognition?). Thus, it is left unclear why other behavioral analyses in this study collapsed across the sex variable.

4. In my opinion, the Discussion section is too long. There are several interesting points made regarding the neurobiological effects that may be contributing to the behavioral findings, but none of these were specifically examined. The authors may want to consolidate this information for brevity and conciseness. Of course, if data are available to speak to any of these proposed changes, it would greatly enhance the interpretability of the neurobiological pathways they propose.

5. The authors make the case that the prodromal period of schizophrenia (preceding the onset in adulthood) justifies their approach in employing phencyclidine treatment during the adolescence. It is not clear from the information provided whether this period of susceptibility includes (or is driven by) dysregulation of NMDA-receptor/glutamatergic signaling that would be relevant to their model. This should be described in more detail in the Introduction. More generally, caveats regarding the phencyclidine model of schizophrenia should be mentioned.

6. How do the authors interpret the reduction in “Center Time” exhibited by saline-saline control animals in the open field study on subsequent testing days? Is it possible that this may reflect an acclimation to the testing environment rather than a gauge of emotional responses in subsequent days of testing? If so, the results should provide a clearer delineation between what data relate to anxiety-like behavior (Day 1) versus habituation processes (Days 2 and 3).

7. Given that nicotine reduces some of the behavioral symptoms of phencyclidine-induced schizophrenia, an intriguing possibility is that nicotine may be used as a form of self-medication during this prodromal phase, but runs the risk of creating dependency to something that is harmful in the long-run. Is there evidence that agonists of nicotinic receptors (varencline, nicotine patch) are effective in treating schizophrenic patients?

Minor suggestions:

1. At times, the use of nicotine and NIC is inconsistent.

2. On several occasions, “Latency” is oddly capitalized in the middle of sentences.

3. “Sensorymotor” (title on Figure 4) is misspelled.

4. I would suggest moving 2.3 Materials paragraph closer to the beginning of the Methods section.

5. I would suggest including information regarding the volume dispensed in systemic injections (e.g., 2.5 mg/kg Flunixin, 10 mg/mL injection for instance).

Reviewer #2: In this manuscript, Dutra-Tavares and colleagues examined the effects of nicotine exposure on schizophrenia-like symptoms during adolescence in a mouse model of schizophrenia. On postnatal day 38, mice were surgically prepared with a minipump that delivered nicotine (24 mg/Kg/day) for seven days. From postanal day 38-52, mice were administered with the NMDA antagonist phencyclidine (PCP) to induce a to induce schizophrenia-like phenotype. To assess the schizophrenia-like symptoms, mice were then tested on the open field, novel object recognition, three-chamber social interaction and the prepulse inhibition tests. The authors conclude that nicotine exposure attenuated PCP-induced hyperlocomotion, locomotor sensitization, and sociability. This is valuable given the strong comorbidity between schizophrenia and tobacco use. The overall implications of this study are also useful as they highlight the fact that nicotine exposure impacts schizophrenia-like symptoms in a PCP animal model of schizophrenia during adolescence. Nevertheless, the following are issues that need to be addressed

Below are the main concerns:

What was the rationale for using the PCP model of schizophrenia versus other models?

What was the rational for testing mice after the end of nicotine exposure versus during nicotine exposure? Would similar results have been obtained? What about the impact of nicotine withdrawal on the magnitude of schizophrenia-like symptoms?

Latency to enter the social side was used as measures of social interaction. Typically, a more direct measure of social interaction such as time in the in the interaction zone is used to look a sociability. The lack of an effect in the percent time in the social side does not provide strong support for the conclusion that there was an effect of PCP on sociability.

Previous work (https://doi.org/10.3389/fpsyt.2013.00038) has shown that adolescent rats require higher nicotine volume to produce similar cotinine levels as adults. Was this considered when selecting the dose of nicotine and method of delivery?

The authors should clarify if the tests were counterbalanced or the rational for the order in which mice were tested.

Reviewer #3: The manuscript by Dutra-Tavares et al examines the interaction between schizophrenia and nicotine during adolescence. The authors use a pharmacological approach to induce schizophrenia in mice followed by a chronic nicotine regimen. The authors conduct behavioral tests (post-nicotine treatment) to assess the effects of phencyclidine and phencyclidine + nicotine. The topic of the manuscript is highly significant, given the heightened tobacco use during adolescence in individuals with mental health disorders such schizophrenia. There is also a need for understanding of the nicotine mechanisms underlying these effects. Although the topic of study merits importance, the current form of the manuscript requires revision. My comments to improve the manuscript are stated below:

Major concerns:

1. The timeline of the experiments is thoughtfully planned. However, the timeline of the experiments requires some clarity, specifically regarding the occurrence of injections, overlapping administration of nicotine pump, and behavioral tests. The manuscript would benefit with a summary timeline of testing order and the post-natal dates.

2. The overall experimental design of this study requires clarity. In some data, only males are used and some data both males and females are reported. Were some tests merged with both male and female data? Or were these data separate cohorts of animals?

3. Throughout the results, the statistical approach used in this study is inconsistent and requires clarity. In some instances (e.g. locomotor activity), several ANOVAs were tested inconsistent with the design approach of the study, increasing the likelihood of false positives. Where proper correction factors applied to multiple comparison tests? Also, the manuscript set the p-value threshold to p=0.1 for an interaction to occur and is unclear if this threshold is correct.

4. Throughout the discussion, much focus is given to systems and circuits of schizophrenia, however, the results of the study do not imply synaptic signaling or changes in neurotransmission. The discussion requires more information regarding similar behavioral findings.

5. A major limitation of the manuscript is the lack of metabolism assessment of nicotine during adolescence. It is possible that the phencyclidine altered pharmacokinetics during adolescence that may have impacted the findings. The authors may wish to consider this possibility on their discussion.

Minor Concerns:

1. Please italicize the Latin phrase “ad libitum” in the methods section.

2. Please expand the diffusion rate of the nicotine pump in the methods sections.

3. What is the rationale of testing this dose of nicotine? Is the dose salt form or free base?

4. In the methods, how was flunixin and Enrofloxacino administered?

5. Please include more detailed information of buffers used to dissolve nicotine and phencyclidine in the methods sections.

6. Body mass findings would benefit as a table of values.

6. PLOS authors have the option to publish the peer review history of their article (what does this mean?). If published, this will include your full peer review and any attached files.

Reviewer #1: No

Reviewer #2: No

Reviewer #3: No

---

## [Author Response · Author response to Decision Letter 0]

13 Aug 2021

The response to reviewers' letter is attached by the end of the PDF file.

---

## [Decision Letter · Decision Letter 1]

15 Sep 2021

Does nicotine exposure during adolescence modify the course of schizophrenia-like symptoms? Behavioral analysis in a phencyclidine-induced mice model

PONE-D-21-15988R1

Dear Dr. Abreu-Villaça,

We’re pleased to inform you that your manuscript has been judged scientifically suitable for publication and will be formally accepted for publication once it meets all outstanding technical requirements.

Kind regards,

Sergio D. Iñiguez, Ph.D.

Academic Editor

PLOS ONE

Additional Editor Comments (optional):

Reviewers' comments:

Reviewer's Responses to Questions

**Comments to the Author**

1. If the authors have adequately addressed your comments raised in a previous round of review and you feel that this manuscript is now acceptable for publication, you may indicate that here to bypass the “Comments to the Author” section, enter your conflict of interest statement in the “Confidential to Editor” section, and submit your "Accept" recommendation.

Reviewer #1: All comments have been addressed

Reviewer #2: All comments have been addressed

Reviewer #3: All comments have been addressed

2. Is the manuscript technically sound, and do the data support the conclusions?

Reviewer #1: (No Response)

Reviewer #2: Yes

Reviewer #3: Yes

3. Has the statistical analysis been performed appropriately and rigorously? 

Reviewer #1: (No Response)

Reviewer #2: Yes

Reviewer #3: Yes

4. Have the authors made all data underlying the findings in their manuscript fully available?

Reviewer #1: (No Response)

Reviewer #2: Yes

Reviewer #3: Yes

5. Is the manuscript presented in an intelligible fashion and written in standard English?

Reviewer #1: (No Response)

Reviewer #2: Yes

Reviewer #3: Yes

6. Review Comments to the Author

Reviewer #1: (No Response)

Reviewer #2: The authors have done a thorough job in addressing the review comments.

Reviewer #3: The authors have given serious consideration to the comments. They have revised their manuscript in a way that addresses each of the comments and concerns. I have no further criticisms.

7. PLOS authors have the option to publish the peer review history of their article (what does this mean?). If published, this will include your full peer review and any attached files.

Reviewer #1: No

Reviewer #2: No

Reviewer #3: No

---

## [Editor Report · Acceptance letter]

20 Sep 2021

PONE-D-21-15988R1 

Does nicotine exposure during adolescence modify the course of schizophrenia-like symptoms? Behavioral analysis in a phencyclidine-induced mice model 

Dear Dr. Abreu-Villaça:

I'm pleased to inform you that your manuscript has been deemed suitable for publication in PLOS ONE. Congratulations! Your manuscript is now with our production department. 

Kind regards, 

on behalf of

Dr. Sergio D. Iñiguez 

Academic Editor

PLOS ONE